# Intratracheal Transplantation of Mesenchymal Stem Cells Attenuates Hyperoxia-Induced Microbial Dysbiosis in the Lungs, Brain, and Gut in Newborn Rats

**DOI:** 10.3390/ijms23126601

**Published:** 2022-06-13

**Authors:** So Yoon Ahn, Dong Kyung Sung, Yun Sil Chang, Won Soon Park

**Affiliations:** 1Department of Pediatrics, Samsung Medical Center, School of Medicine, Sungkyunkwan University, Seoul 06351, Korea; soyoon.ahn@samsung.com (S.Y.A.); cys.chang@samsung.com (Y.S.C.); 2Cell and Gene Therapy Institute, Samsung Medical Center, Seoul 06351, Korea; dbible@hanmail.net; 3Department of Health Sciences and Technology, Samsung Advanced Institute for Health Sciences & Technology SAIHST, Sungkyunkwan University, Seoul 06351, Korea

**Keywords:** microbiota, stem cells, dysbiosis

## Abstract

We attempted to determine whether intratracheal (IT) transplantation of mesenchymal stem cells (MSCs) could simultaneously attenuate hyperoxia-induced lung injuries and microbial dysbiosis of the lungs, brain, and gut in newborn rats. Newborn rats were exposed to hyperoxia (90% oxygen) for 14 days. Human umbilical cord blood-derived MSCs (5 × 10^5^) were transplanted via the IT route on postnatal day (P) five. At P14, the lungs were harvested for histological, biochemical, and microbiome analyses. Bacterial 16S ribosomal RNA genes from the lungs, brain, and large intestine were amplified, pyrosequenced, and analyzed. IT transplantation of MSCs simultaneously attenuated hyperoxia-induced lung inflammation and the ensuing injuries, as well as the dysbiosis of the lungs, brain, and gut. In correlation analyses, lung interleukin-6 (IL-6) levels were significantly positively correlated with the abundance of Proteobacteria in the lungs, brain, and gut, and it was significantly inversely correlated with the abundance of Firmicutes in the gut and lungs and that of Bacteroidetes in the lungs. In conclusion, microbial dysbiosis in the lungs, brain, and gut does not cause but is caused by hyperoxic lung inflammation and ensuing injuries, and IT transplantation of MSCs attenuates dysbiosis in the lungs, brain, and gut, primarily by their anti-oxidative and anti-inflammatory effects.

## 1. Introduction

Despite recent advances in neonatal intensive care medicine, bronchopulmonary dysplasia (BPD), a chronic disease of premature infants receiving oxygen therapy and mechanical ventilation, remains a major cause of mortality and long-term pulmonary and neurodevelopmental morbidities, with few effective treatment options [1]. Although our recent preclinical [2,3,4,5,6] and clinical studies [7,8,9] suggest that mesenchymal stem cell (MSC) transplantation may be a promising novel, safe, and effective therapeutic option for BPD, a better understanding of the pathogenesis of BPD and the protective mechanism of MSC transplantation is essential for its successful translation into clinical practice in the near future.

Compared to traditional culture methods, Microbiome analysis with sequencing of 16S ribosomal RNA genes has facilitated the detection of a rich microbiome in the lungs and brain, which were previously thought to be sterile tissues [10]. Alterations in airway microbial communities of mechanically ventilated preterm infants have been associated with BPD progression and severity [11,12]. Moreover, the lung and gut microbiome are closely associated (gut–lung axis), indicating that gut dysbiosis may also contribute to lung injury, leading to BPD in premature infants [13,14].

In a recent schizophrenia (SZ) animal study using metabotropic glutamate receptor 5 knockout mice, the microbiome phenotype observed in mGlu5 KO mice is to some extent in line with reports of gut dysbiosis in SZ patients. Thus, this animal model seems to provide a tool to investigate the link between microbial dysbiosis and schizophrenia symptoms via clarifying the extent of microbiota–gut–brain axis dysfunction in this model [15]. Like SZ, in BPD little is known about the effect of BPD on the lungs, brain, and gut microbiome and whether these alterations may be modulated by stem cell transplantation. Thus, we attempted to determine whether intratracheal (IT) transplantation of MSCs could simultaneously attenuate hyperoxia-induced lung injuries and microbial dysbiosis of the lungs, brain, and gut in newborn rats. Furthermore, we determined whether the extent of microbial dysbiosis in the lungs, brain, and gut is closely associated with the severity of lung inflammation and whether a close link exists among the lungs, brain, and gut microbial dysbiosis.

## 2. Results

### 2.1. Body Weight and Lung Morphometry at P14

Newborn Sprague–Dawley (SD) rats were randomly assigned to either the normoxia control group (NC), hyperoxia control group (HC), or hyperoxia with human umbilical cord blood (UCB)-derived MSCs transplantation group (HM). Newborn SD rats were exposed to normoxia or hyperoxia (90% oxygen) for 14 days. At postnatal day five, HM group rats received MSCs intratracheally. At postnatal day 14, brain, lungs, and gut tissues were harvested after sacrifice.

Although birth weight did not differ significantly between the study groups, body weight at P14 in HC rats was significantly lower than that in NC rats. Moreover, the apparent increase in the body weight in the HM group compared to that in the HC group was not statistically significant (Figure 1A). Representative light microscopic photomicrographs showing the histopathological differences between the study groups are shown in Figure 1C. Compared with the small and uniform alveoli in NC, fewer, larger, heterogeneously sized, and corrugated alveoli, indicative of impaired alveolarization, were observed in HC, and hyperoxia-induced impaired alveolar development was attenuated in the HM group. In morphometric analysis, the mean linear intercept (MLI) indicating alveolar size was significantly higher in the HC group than in the NC group, and this hyperoxia-induced increase in MLI was significantly attenuated in the HM group (Figure 1B).

### 2.2. Changes in the Number of TUNEL- and ED-1-Positive Cells and Levels of Angiogenic and Inflammatory Markers

In HC rat lungs at P14, the number of ectodermal dysplasia (ED)-1- and TUNEL-positive cells and level of inflammatory cytokines such as interleukin (IL)-1α, IL-1β, IL-6, and TNF-α were significantly increased. Moreover, the light intensity of von Willebrand factor (vWF), a marker of angiogenesis, was significantly decreased in the HC group compared to the NC group (Figure 2). These hyperoxia-induced detrimental effects were significantly attenuated by MSC transplantation in HM rats. The hyperoxia-induced increase in gut oxidative stress compared to that in the NC group was significantly attenuated in HM rats Appendix A.

### 2.3. Relative Abundance of Microbial Taxa

Figure 3C shows a representative bar graph depicting the relative abundance of the lung, brain, and gut microbiota using 16S ribosomal RNA gene sequencing in each group. The proportion of microbiota varied according to the host tissue site, and the main recovered phyla were Proteobacteria in the lungs and brain and Firmicutes in the gut. While the Shannon and Simpson diversity indices were not significantly different between the study groups (Figure 3B), the relative abundance of Proteobacteria in the lungs, brain, and gut was significantly increased. Furthermore, the relative abundance of Firmicutes in the gut and lungs and that of Actinobacteria in the brain were significantly decreased in the HC group compared to the NC group.

The hyperoxia-induced increase in the abundance of lung Proteobacteria and decrease in the abundance of gut Firmicutes were significantly attenuated in the HM group (Table 1).

At the genus level, the relative abundance of gut Lactobacillus was significantly decreased and that of Escherichia-Shigella was significantly increased in the HC group compared to the NC group, and the hyperoxia-induced gut microbial dysbiosis was significantly attenuated in the HM group (Table 2).

### 2.4. Correlation Analyses of Lung Inflammation and Lung, Brain, and Gut Microbiota

In correlation analyses, lung IL-6 levels were significantly positively correlated with the abundance of Proteobacteria in the lungs (*R*^2^ = 0.52, *p* < 0.01), brain (*R*^2^ = 0.37, *p* < 0.05), and gut (*R*^2^ = 0.47, *p* < 0.01) and were significantly negatively correlated with the abundance of Firmicutes in the gut (*R*^2^ = 0.59, *p* < 0.01) and lungs (*R*^2^ = 0.51, *p* < 0.01), and that of Bacteroidetes in the lungs (*R*^2^ = 0.30, *p* < 0.05) (Figure 4). In addition to IL-6, well known pulmonary inflammatory cytokines such as lung TNF-α and IL-1α/β levels also showed similar trends and patterns with IL-6 in correlation analysis with lung, brain, and gut microbiota.

## 3. Discussion

Previous studies have not determined whether MSC transplantation can simultaneously attenuate both hyperoxia-induced lung injuries and microbial dysbiosis of the lungs, brain, and gut in newborn rats. In the present study, we detected bacterial microbiota using the 16S rRNA screening method even in tissues generally thought to be sterile, such as the lungs and brain [10]. Different parts of the body have different proportions of bacteria, with Proteobacteria and Firmicutes being the predominant phyla in the normal lungs/brain and large intestine, respectively [16,17]. Considering the variable normal microbiota composition in the different GI tracts, with predominant anaerobic phyla in the anoxic colon and predominant facultative anaerobic phyla in the oxygenated small intestine [18,19,20], and the key role of Proteobacteria in preparing the tissue for colonization by the strict anaerobes necessary for healthy tissue function by consuming oxygen [21,22] the oxygenation status of each body part might be a major determinant of its bacterial proportion. Further studies are necessary to clarify the complex mechanisms of maintaining normal microbiota homeostasis in each body part.

The major findings of the present study were the significant attenuation of hyperoxia-induced lung inflammation and the ensuing impaired alveolarization and angiogenesis. The IT transplantation of MSCs simultaneously attenuated the hyperoxia-induced increase in the relative abundance of Proteobacteria in the lungs, brain, and gut, and a concomitant decrease in the relative abundance of Firmicutes in the lungs and gut in newborn rats. These findings suggest that microbial dysbiosis in the gut, lungs, and brain is not causal but occurs in response to hyperoxia-induced inflammation leading to BPD; therefore, IT transplantation of MSCs simultaneously attenuates hyperoxia-induced lung injuries and microbial dysbiosis in the lungs, brain, and gut, primarily by their anti-oxidative and anti-inflammatory effects [6,23].

In concordance with our data, a significantly increased relative abundance of Proteobacteria and a concomitant decrease in the relative abundance of Firmicutes were also observed in the airway microbiome of preterm infants with BPD [11,12,24], fecal microbiome of preterm infants with NEC [25], the adult lung microbiome of exacerbated chronic obstructive pulmonary disease [26] and the gut microbiome of inflammatory bowel disorder [18]. These findings suggest that the increased abundance of the facultative anaerobic Proteobacteria along with the decreased abundance of the obligate anaerobic Firmicutes can be a universal microbial diagnostic signature of dysbiosis in many inflammatory and allergic disorders, including ‘oxygen free radical disorders of neonatology’, such as BPD, NEC, and periventricular leukomalacia, regardless of the underlying disorder [27,28].

In our previous phase 1 and phase 2 human clinical trial of MSCs transplantation for premature infants, lung inflammatory cytokine levels of IL-6 and TNF-α showed significant difference in tracheal aspirate fluid between the placebo control group and MSC group [8,9]. Based on this data, we performed correlation analysis using pulmonary IL-6 level as a severity marker of pulmonary inflammation in this present study. Besides IL-6, other pulmonary cytokine levels of TNF-α and IL-1α/β also showed similar trends and pattern with IL-6 in correlation analysis. Though we observed microbial changes in lung, gut and brain according to the lung IL-6 levels, it is not sufficient to suggest the unique role of IL-6 in controlling microbial composition of different organs. When substantial data about possible role of IL-6 in human gut microbiota and microbiota associated lung-gut axis [29,30,31] are considered, this correlation of lung IL-6 and microbial composition in lung, gut, and brain strongly requires further investigation of the role of IL-6 in regulating microbial environments.

In our previous and present studies [2,3,4,5,6,8,23], IT transplantation of MSCs simultaneously attenuated the hyperoxia-induced increase in oxidative stress and inflammation and the resultant microbial dysbiosis in the lungs, brain, and gut. Moreover, in correlation analyses, the abundance of Proteobacteria in the lungs, brain, and gut was significantly positively correlated with the extent of lung inflammation, and the abundance of Firmicutes in the lungs and gut was significantly negatively correlated with the extent of lung inflammation. Collectively, these data suggest that increased oxygen availability during inflammation in the lungs, gut, and brain drives a dysbiotic expansion of the facultative anaerobic Proteobacteria and the concomitant shrinkage of obligate anaerobic Firmicutes [20,21]. These observations suggest that attenuating oxidative stress and the ensuing inflammation by IT transplantation of MSCs may therapeutically rebalance the hyperoxia-induced dysbiosis in the lungs, brain, and gut, which can be beneficial for healing hyperoxic lung, brain, and gut injuries.

Our study has several limitations. While the detrimental effects of profound hyperoxia (target FiO_2_ 90%) on lungs, brain, and gut biology are well established [5,23], the significance of mild hyperoxia in routine clinical care of premature infants is less well established. Although our hyperoxic newborn rat pup model simulated the clinical conditions of BPD, the histology recapitulated the key features of human BPD in premature infants, and the microbial taxonomic alterations coincided with the human data, there might be unnoticed important anatomical, immunologic, and microbiologic differences between newborn rats and human preterm infants that would necessitate further validation. Lastly, although our data do not support the causal effects of microbial dysbiosis on hyperoxia-induced lung injuries, further studies are necessary to determine whether supplementation with probiotics such as Lactobacillus could be beneficial for hyperoxic lung injuries [14].

## 4. Materials and Methods

### 4.1. Animal Model

All animal procedures were approved by the Institutional Animal Care and Use Committee of the Samsung Biomedical Research Institute, Seoul, Korea, and were performed in an AAALAC-accredited specific pathogen-free facility. These procedures were performed in accordance with our institutional guidelines and the National Institutes of Health Guidelines for Laboratory Animal Care. Timed-pregnant Sprague–Dawley (SD) rats (Orient Co., Seoul, Korea) spontaneously delivered newborn rat pups, as described previously [2,3,4,5,6,8]. Newborn SD rats reared with their dams in the standard cage, a 50 L Plexiglas chamber, were used in this study. Dam rats were maintained in an alternating 12-h light/dark cycle with constant room humidity and temperature. The condition of the rat pups was assessed and monitored every week, twice daily, especially for the 14 days after hyperoxia exposure. In this study, we used humane endpoint as the earliest indicator of pain or distress in an animal experiment that could be used to avoid or limit pain and distress by taking actions such as humane euthanasia. For the humane endpoint, an operationally defined scoring system was approved by the IACUC. Total scores ≥ 5 or a score of 3 in any single category were arbitrarily defined as the humane endpoint. Humane endpoints consisted of body weight growth (1, slower growth than normal rats; 2, growth arrest; 3, weight loss), responsiveness (1, delayed but appropriate response; 2, delayed and null response; 3, no response), and appearance (1, rough hair coat; 2, porphyrin staining; 3, sustained abnormal posture or dilated pupil). Throughout the experimental period, no rat pups reached the humane endpoint. Newborn rats were randomly allocated to three experimental groups: normoxia control group (NC, *n* = 6), hyperoxia control group (HC, *n* = 10), and hyperoxia with IT human umbilical cord blood (UCB)-derived MSCs transplantation group (HM, *n* = 10). Normoxic rats were raised in room air, whereas hyperoxic rats were raised in hyperoxic chambers containing 90% oxygen from birth until postnatal day (P) 14. At P5, The IT administration of MSCs (5 × 10^5^ cells/50 µL) was performed under inhalation anesthesia induced with a mixture of halothane and 2:1 nitrous oxide: oxygen, and all efforts were made to minimize suffering. At P14, rat pups were sacrificed under deep pentobarbital anesthesia (60 mg/kg, IP), and lung tissue was harvested for morphometric, biochemical (NC: *n* = 6, HC: *n* = 10, HM: *n* = 10), and microbiome analyses (NC: *n* = 3, HC: *n* = 5, HM: *n* = 7).

### 4.2. Preparation of MSCs and Transplantation

Human UCB-derived MSCs obtained from a single donor and manufactured in strict compliance with the good manufacturing process at passage 6 (MEDIPOST Co., Ltd., Seoul, Korea) were used in the present study [2,3,4,5,6,8,32,33]. The characterization [34], differentiation potential [6,35], immunophenotypic results [6] and karyotypic stability up to passage 11 of MSCs [3] have been reported in our previous studies. A single dose of human UCB-derived MSCs (5 × 10^5^ cells in 0.05 mL of normal saline) for HM or an equal volume of normal saline for HC was administered intratracheally at P5.

### 4.3. Morphometric Analysis

Paraffin sections (4 μm thick) were stained with hematoxylin and eosin. Two sections were randomly chosen per rat, and three random microscopic fields of the distal lung were analyzed for each section. The level of alveolarization was determined by measuring the mean linear intercept (MLI) and mean alveolar volume (MAV) as previously described [6].

### 4.4. TUNEL Staining

Apoptotic cells were stained using the ApopTag Fluorescein Apoptosis Detection Kit (Chemicon, Temecula, CA, USA) according to the manufacturer’s protocol. Slides were mounted in VECTASHIELD mounting medium containing DAPI and visualized using fluorescence microscopy. The number of terminal deoxynucleotidyl transferase dUTP nick end labeling (TUNEL)-positive cells was determined in 10 non-overlapping random fields per animal in a blinded manner.

### 4.5. Quantification of ED-1-Positive Cells

Immunohistochemical analysis of reactive microglia (ED-1) was performed on deparaffinized 4-μm lung sections. The specimens were placed in a solution containing 0.1% (*v*/*v*) Triton X-100 and 0.5% (*v*/*v*) bovine serum albumin (BSA) in phosphate-buffered saline (PBS) and incubated with primary anti-monocyte/macrophage antibodies (1:100; anti-CD68 ED-1 mouse monoclonal, Chemicon, Millipore, MA, USA). The sections were then stained with fluorescein isothiocyanate (FITC)-conjugated polyclonal rabbit anti-mouse immunoglobulins (1:1000, 1:200) for 1 h at room temperature. VECTASHIELD mounting medium with DAPI (Vector Laboratories) was used to counterstain nuclei. ED-1-positive cells were counted manually and averaged per HPF in a single animal. Two random sections per animal were evaluated in a blinded manner.

### 4.6. ELISA Assay

Frozen lung samples were homogenized in lysis buffer, and lysates were clarified by centrifugation at 13,000× *g* for 20 min at 4 °C to remove cellular debris. The protein content of the supernatants was quantified by the Bradford method, using bovine serum albumin as the standard. Levels of lung macrophage inflammatory protein-1 α (MIP-1α), tumor necrosis factor-α (TNF-α), and interleukin-6 (IL-6) were measured using a Milliplex MAP ELISA Kit, according to the manufacturer’s protocol (Millipore, Billerica, MA, USA). Levels of vascular endothelial growth factor (VEGF) were measured using the R&D Rat VEGF Quantikine ELISA kit according to the manufacturer’s protocol (R&D Systems, Minneapolis, MN, USA).

### 4.7. Oxidative Stress Analysis

To evaluate oxidative stress, we quantified Malondialdehyde (MDA) in intestine. Samples of intestine were homogenized with normal saline, and centrifuged at 4000× *g* for 15 min at 4 °C. The supernatants were transferred into fresh tubes for evaluation of the MDA levels. The level of malondialdehyde (MDA), a marker of oxidative stress, was evaluated in duplicate in cell lysates using the MDA assay kit containing thiobarbituric acid-reactive substances (Cell Biolabs, San Diego, CA, USA) according to the manufacturer’s protocol.

### 4.8. Barcoded Deep 454-Pyrosequencing of 16S rRNA Gene Amplicon

All tissue preparation procedures for the various analyses were performed only on the surviving animals at P14. The animals were anesthetized with sodium pentobarbital (100 mg/kg). The lungs were exposed by thoracotomy and transcardial perfusion with ice-cold PBS. The whole brain, lungs, and large intestine were extracted and rapidly snap-frozen, stored at −80 °C, and homogenized shortly before analyses. The 16S rRNA genes of 15 samples were amplified by nested PCR using the Gammaproteobacteria primer set 395f (5′-CMA TGC CGC GTG TGT GAA-3′) and 871r (5′-ACT CCC CAG GCG GTC DAC TTA-3). The PCR reaction mixture (20 μL) contained 5 × Taq-&GO Ready-to-use PCR Mix (MP Biomedicals, Heidelberg, Germany), 0.25 μM of each primer, 25 mM MgCl_2,_ and 1 μL of template DNA (96 °C, 4 min; 32 cycles of 96 °C, 1 min; 57 °C, 1 min; 74 °C, 1 min; and final elongation at 74 °C, 10 min). In the second PCR, 1 μL of the amplicon (1:10 diluted phyllosphere and 1:100 diluted rhizosphere-derived PCR products) was used. The 16S rRNA gene sequences were amplified using the forward primer Unibac-II-515f (5-GTG CCA GCA GCC GC-3) containing the 454-pyrosequencing adaptors and the reverse primer Gamma871r_454 (5-CTA TGC GCC TTG CCA GCC CGC TCA GAC TCC CCA GGC GGT CDA CTT A-3′) [36]. The reaction mixture for the second PCR (30 μL) contained 5 × Taq-&GO Ready-to-use PCR Mix, 0.25 μM of each primer (96 °C, 4 min; 32 cycles of 96 °C, 1 min; 66 °C, 1 min; 74 °C, 1 min; and final elongation at 74 °C, 10 min). The PCR products were purified using the Wizard SV Gel and PCR Clean-Up System (Promega, Madison, WI, USA). The technical replicates per sample were pooled, and the partial 16S rRNA gene fragments were sequenced using a 454 Roche GS FLX (MWG Eurofins, Ebersberg, Germany) and 454 Roche GS FLX Titanium pyrosequencer (Macrogen Korea, South Korea). 

### 4.9. DNA Sequence Analysis and Taxonomical Identification

Sequences were analyzed using Qiime software version 6.0 [37]. Replicates from sequencing of each treatment and habitat were bioinformatically pooled during Qiime analysis for data evaluation. MID, primer, and adapter sequences were removed, length filtered (≥350 nt), quality filtered (score: 50), denoised, chloroplast removed, and singletons adjusted. The cutoff level was set to 97% sequence identity. Chimeras were detected using the Chimera Slayer and then removed. To compute alpha and beta diversities, the dataset was normalized to 5920 reads per sample. Ring charts were plotted using the Krona software package version 2.2 [38] and the profile network was constructed using Cytoscape version 3.0.2 [39]. Statistical tests based on the operational taxonomic units (OTUs) table were performed using the nonparametric ANOVA Kruskal–Wallis test. This test is functionally an expansion of ANOVA for cases in which the sample means are unequal, and the distribution is not normal.

### 4.10. Statistical Analysis

Data are expressed as the mean ± standard deviation. The Mann–Whitney U test was used to compare continuous variables between groups, and Spearman’s correlation was used to measure the association between continuous variables. Statistical significance was set at *p* < 0.05. Data were analyzed using SPSS software (version 17.0; SPSS Institute, Chicago, IL, USA).

## 5. Conclusions

IT transplantation of MSCs simultaneously attenuated hyperoxia-induced inflammation and the ensuing lung injuries and microbial dysbiosis of the lungs, brain, and gut in newborn rats. The shift in bacterial communities from obligate to facultative anaerobes was significantly correlated with the extent of lung inflammation. These findings suggest that microbial dysbiosis in the lungs, brain, and gut does not cause but results from hyperoxia-induced inflammation leading to BPD, and that IT transplantation attenuates hyperoxia-induced dysbiosis mainly by its anti-oxidative and anti-inflammatory effects.

## Figures and Tables

**Figure 1 ijms-23-06601-f001:**
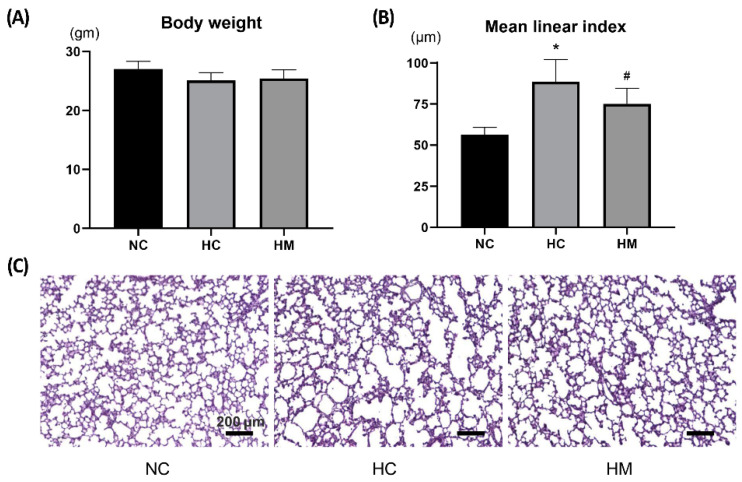
The body weight and degree of alveolarization of P14 rat lungs exposed to prolonged hyperoxia in vivo. Body weight (**A**) and degrees of alveolarization were assessed by the mean linear intercept of the lung sections (**B**) of P14 rats. (**C**) Representative optical microscopy photomicrographs of lung sections stained with hematoxylin and eosin (scale bar = 200 μm). Data are presented as the mean ± standard deviation. Abbreviations: NC, normoxia control; HC, hyperoxia control; HM, hyperoxia with transplantation of human UCB-MSCs. *, *p* < 0.05 compared with the NC group; #, *p* < 0.05 compared with the HC group.

**Figure 2 ijms-23-06601-f002:**
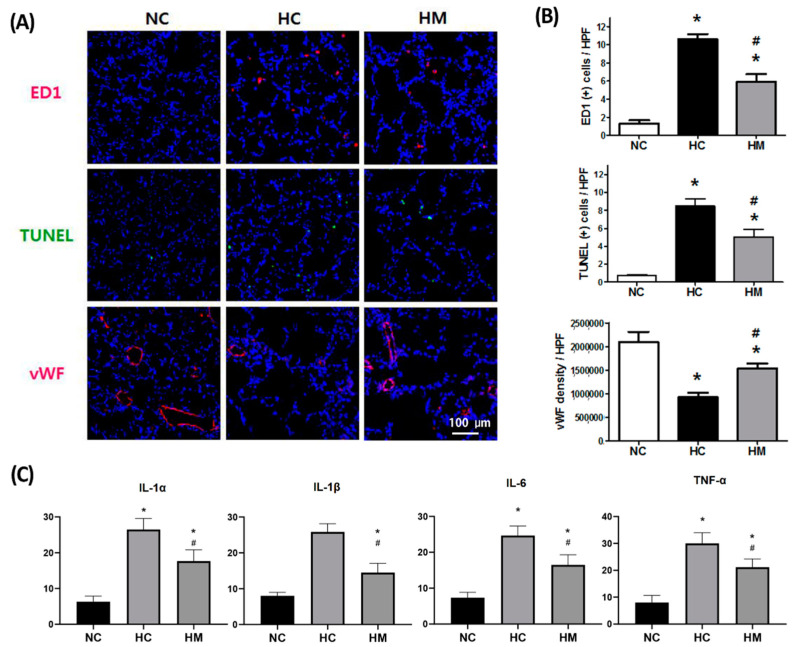
Evidence of inflammation, cell death, and pulmonary angiogenesis in P14 rat pups. (**A**) Representative immunofluorescence photomicrographs of ED-1-positive cells, TUNEL-positive cells, and von Willebrand factor (vWF) staining in the lungs of P14 rats. ED-1-positive alveolar macrophages and vWF were labeled with the fluorescent marker (red), and TUNEL was labeled with FITC (green). Nuclei were labeled with 4′,6-diamidino-2-phenylindole (DAPI, blue) (scale bars = 100 μm). (**B**) Numbers of observed TUNEL- and ED-1-positive cells and mean light signal intensity of vWF immunofluorescence staining per high power field in lung sections from P14 rats. (**C**) Levels of inflammatory cytokines of IL-1α, IL-1β, IL-6, and TNF-α in the lungs of P14 rats. Data are presented as the mean ± standard deviation. Abbreviations: NC, normoxia control; HC, hyperoxia control; HM, hyperoxia with transplantation of human UCB-MSCs. *, *p* < 0.05 compared with the NC group; #, *p* < 0.05 compared with the HC group.

**Figure 3 ijms-23-06601-f003:**
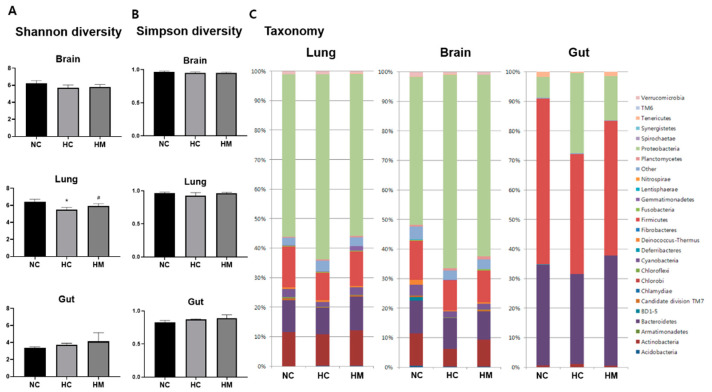
Microbiome diversity indices and relative abundances of microbial taxonomies in the brain, lungs, and gut. (**A**) Shannon and (**B**) Simpson diversity indices, and (**C**) relative abundances of microbial taxonomies in the brain, lungs, and gut. Data are presented as the mean ± standard deviation. Abbreviations: NC, normoxia control; HC, hyperoxia control; HM, hyperoxia with transplantation of human UCB-MSCs. *, *p* < 0.05 compared with the NC group; #, *p* < 0.05 compared with the HC group.

**Figure 4 ijms-23-06601-f004:**
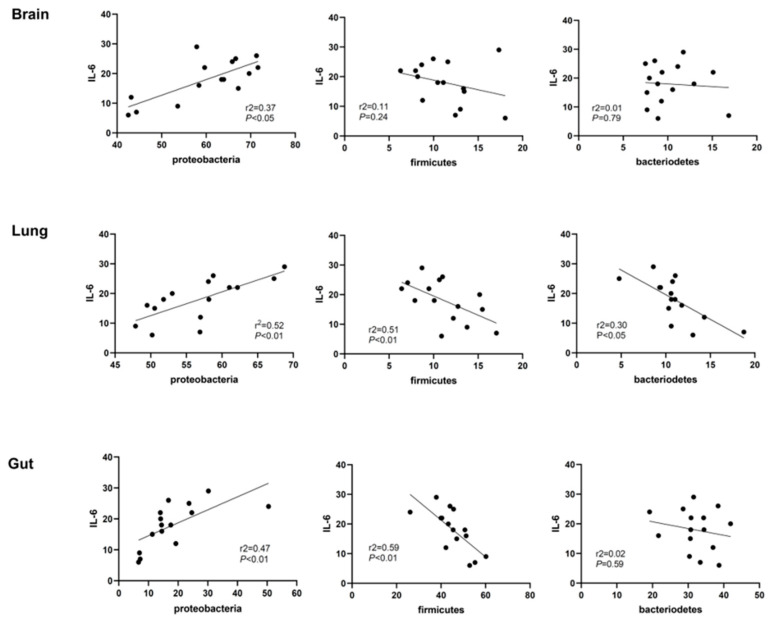
Association between the abundance of microbiome and IL-6 expression in the lungs. Correlation between the IL-6 levels in the lung tissue homogenates and the abundance of microbiomes such as Proteobacteria, Firmicutes, and Bacteroidetes in the brain, lungs, and gut.

**Table 1 ijms-23-06601-t001:** Relative abundance of phyla.

**Lung Phylum**	**NC**	**HC**	**HM**
Proteobacteria	51.66 ± 4.69	62.69 ± 4.42 *	52.69 ± 3.63 ^#^
Firmicutes	13.9 ± 3.08	8.89 ± 1.87 *	11.78 ± 2.97
Bacteroidetes	14.15 ± 4.19	8.99 ± 2.27	11.81 ± 1.67
Actinobacteria	11.5 ± 2.75	10.54 ± 3.74	11.66 ± 3.01
Cyanobacteria	2.7 ± 1.64	1.69 ± 0.87	2.02 ± 0.99
**Brain Phylum**	**NC**	**HC**	**HM**
Proteobacteria	48.16 ± 8.24	65.49 ± 5.74 *	61.53 ± 8.82
Firmicutes	14.5 ± 3.09	10.30 ± 3.89	10.75 ± 2.06
Bacteroidetes	11.15 ± 4.97	10.56 ± 2.73	9.63 ± 1.8
Actinobacteria	13.01 ± 4.26	5.94 ± 2.66 *	9.12 ± 8.08
Cyanobacteria	3.23 ± 0.5	2.07 ± 0.27 *	1.86 ± 0.32
**Gut Phylum**	**NC**	**HC**	**HM**
Firmicutes	56.11 ± 0.36	39.0 ± 6.91 *	46.59 ± 3.45 *^,#^
Bacteroidetes	34.16 ± 4.2	30.48 ± 6.48	32.73 ± 6.27
Proteobacteria	6.96 ± 0.27	26.63 ± 13.06 *	15.20 ± 2.8
Tenericutes	1.66 ± 1.5	1.08 ± 1.7	0.95 ± 1.75
Actinobacteria	0.57 ± 0.3	0.87 ± 1.15	1.61 ± 0.72

Group abbreviations are as follows: NC, normoxia control; HC, hyperoxia control; HM, hyperoxia with transplantation of human UCB-MSCs. *, *p* < 0.05 compared with the NC group; ^#^, *p* < 0.05 compared with the HC group.

**Table 2 ijms-23-06601-t002:** Relative abundance of genus.

**Lung Phylum**	**Lung Genus**	**NC**	**HC**	**HM**
Proteobacteria	Methylophaga	10.51 ± 4.65	12.85 ± 2.79	7.07 ± 3.03 ^#^
Proteobacteria	Sphingomonas	8.44 ± 0.89	8.80 ± 1.56	6.71 ± 2.29
Actinomycetota	Priopionibacterium	6.11 ± 0.88	5.30 ± 3.21	7.05 ± 4.11
Proteobacteria	Aeromonas	5.21 ± 1.47	3.41 ± 0.72	3.83 ± 1.76
Bacillota	Streoptococcus	4.33 ± 0.41	3.86 ± 1.33	3.89 ± 1.97
**Brain Phylum**	**Brain Genus**	**NC**	**HC**	**HM**
Proteobacteria	Methylophaga	7.99 ± 0.70	15.14 ± 5.72 *	15.54 ± 5.21
Proteobacteria	Sphingomonas	6.76 ± 0.74	9.67 ± 2.15	8.45 ± 1.52
Bacillota	Streoptococcus	4.81 ± 2.35	5.29 ± 2.72	5.55 ± 1.50
Proteobacteria	Aeromonas	3.16 ± 0.13	4.90 ± 1.12 *	4.04 ± 1.67
Actinomycetota	Priopionibacterium	5.44 ± 1.45	2.89 ± 1.43 *	4.72 ± 4.70
**Gut Phylum**	**Gut Genus**	**NC**	**HC**	**HM**
Firmicutes	Lactobacillus	43.78 ± 8.29	22.27 ± 9.58 *	34.51 ± 4.61 ^#^
Bacteroidetes	Bacteroides	21.52 ± 4.49	14.22 ± 6.39	17.50 ± 10.03
Bacteroidetes	Parabacteroides	12.61 ± 4.02	15.78 ± 2.02	10.57 ± 6.97
Proteobacteria	Escherichia-Shigella	4.00 ± 0.82	19.59 ± 6.18 *	8.32 ± 6.50 ^#^
Unknown	Unknown	4.90 ± 4.24	7.75 ± 4.59	2.73 ± 2.48

Group abbreviations are as follows: NC, normoxia control; HC, hyperoxia control; HM, hyperoxia with transplantation of human UCB-MSCs. *, *p* < 0.05 compared with the NC group; ^#^, *p* < 0.05 compared with the HC group.

## Data Availability

The datasets generated and analysed during this current study are available from the corresponding author on reasonable request.

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
