# Peer review of "Intratracheal Transplantation of Mesenchymal Stem Cells Attenuates Hyperoxia-Induced Microbial Dysbiosis in the Lungs, Brain, and Gut in Newborn Rats"

_ijms, 2022, doi:10.3390/ijms23126601_

Round 1
Reviewer 1 Report
The manuscript addresses a very interesting point about the hyperoxia-induced microbial dysbiosis in the lungs, brain and gut in newborn rats. The manuscript is well written. The authors could expand an introduction a bit to address if there are any modeling studies completed on identifying the microbial dysbiosis.
Author Response
-> We appreciate to reviewer’s helpful comment. We added study of disease animal model for investigating microbial dysbiosis in the introduction (Page 2, line47-52) as below:
In recent schizophrenia (SZ) animal study using metabotropic glutamate receptor 5 knockout mouse, the microbiome phenotype observed in mGlu5 KO mice is to some extent in line with reports of gut dysbiosis in SZ patients. Thus, this animal model seems to provide a tool to investigate the link between microbial dysbiosis and schizophrenia symptoms via clarifying the extent of microbiota–gut–brain axis dysfunction in this model (reference #15).

Reviewer 2 Report
In the article “Intratracheal Transplantation of Mesenchymal Stem Cells At- 2 tenuates Hyperoxia-Induced Microbial Dysbiosis in the Lungs, Brain, and Gut in Newborn Rats” the authors demonstrate that intratracheal transplantation of mesenchymal stem cells reduced hyperoxia-induced lung inflammation and related injuries, as well as the dysbiosis of the lungs, brain, and gut
Moreover, lung interleukin-6 levels were correlated with the abundance of specific pro- and anti-inflammatory bacteria in the organs concluding that microbial dysbiosis is caused by hyperoxic lung inflammation and ensuing injuries, and that intratracheal transplantation of mesenchymal stem cells can attenuate dysbiosis trough anti-oxidative and antiinflammatory mechanisms.
The impact of these pre-clinical results is high considering that mesenchymal stem cell transplantation may be a promising novel, safe, and effective therapeutic option for bronchopulmonary dysplasia in future clinical practice.
Methods are appropriate and well described.
Why the authors performed correlation analyses exclusively for lung IL-6 levels?
How the authors can demonstrate the correlation among lung IL-6 and microbial composition of different organs? Some specific markers to clearly demonstrate the connection and related pathways should be included in the study to validate these interesting observations.
Did the authors performed behavioral tests to verify the positive or negative impact of these invasive procedure on cognition?
Why the authors used the standard error of the mean and not the standard deviation?
The expression of carbonylated proteins (supplementary material) is not convincing. Bands are too faint to generate the densitometry shown.
